# Genetic Algorithm Optimization of Beams in Terms of Maximizing Gaps between Adjacent Frequencies

**DOI:** 10.3390/ma16144963

**Published:** 2023-07-12

**Authors:** Łukasz Domagalski, Izabela Kowalczyk

**Affiliations:** Department of Structural Mechanics, Lodz University of Technology, Politechniki 6, 93-590 Lodz, Poland; izabela.kowalczyk@dokt.p.lodz.pl

**Keywords:** genetic algorithms, optimization, band gaps, beams, eigenvalue problem, dynamic analysis, finite element method

## Abstract

The aim of this paper is to optimize the thickness variation function of simply supported and cantilever beams, in terms of maximizing gaps between chosen neighboring frequencies, and to analyze the obtained results. The optimization results are examined in terms of achieving the objective function (related to eigenvalue problems), but also in terms of their dynamic stiffness (forced vibrations excited by a point harmonic load). In the optimization process, a genetic algorithm was used. Problems related to structural dynamics were solved by FEM implementation into the algorithm. Sample results were presented, and the developed algorithm was analyzed in terms of the results convergence by examining several variable parameters. The authors demonstrated the validity of applying the described optimization tool to the presented problems. Conclusions were drawn regarding the correlation between stiffness and mass distribution in the optimized beams and the natural frequency modes in terms of which they were optimized.

## 1. Introduction

This paper presents an optimization of the thickness variation function of simply supported and cantilever beams, in terms of maximizing gaps between their chosen neighboring frequencies. The method of optimization—sizing optimization—is a method of modifying dimensions or sizes of components in order to obtain the best structural performance. The beams are made of a homogeneous material. Analyzed elements are considered in the framework of the Euler–Bernoulli theory.

There are a number of studies on the band-gap behavior of structural elements. This phenomenon is of significant importance in various engineering applications. By controlling band-gap characteristics, it is possible to alter the mechanical and vibrational properties of a structure, leading to improved performance and functionality. For example, in the field of materials engineering, the design of band gaps has become an essential tool in developing materials with specific properties, such as phononic crystals (PnCs) and metamaterials [1,2,3]. In the field of civil engineering, the band-gap characteristics of structural elements such as beams or plates are related to their dynamic response. Controlling their occurrence can affect the acoustic and vibrational insulation of the structure, which is crucial for the comfort and safety of occupants. It becomes particularly significant in industrial buildings equipped with vibrating devices and machinery. By managing their dynamic properties, resonance risk can be minimized and structural fatigue can be reduced. For similar reasons, it is essential in bridge construction, where vibrations can be generated by human activities and vehicular traffic. Therefore, the design and analysis of structural elements require a deep understanding of their band-gap behavior, so it can lead to the development of lightweight, high-strength, and energy-efficient structures.

So far, many researchers have addressed the issue of optimization and analysis of structures with gaps between frequencies. Some of the first publications in this area were presented by Olhoff et al. in 1970s and 1980s [4,5,6]. The aim of the first of these was to find a transversely vibrating beam that yielded the maximum value of a higher natural frequency of specified order. The authors introduced a simple method of obtaining solutions corresponding to any higher value of the specified order using scaled optimal beam elements. In the second work, authors determined the optimal solution of a transversely vibrating, thin elastic beam or rotating shaft, which maximized the difference between two adjacent natural frequencies. The topic of vibrating beams or rotating shafts was also continued in the last of the mentioned papers. In all three works, the solution was derived by variational analysis and solved numerically by an iteration procedure based on finite difference discretization. The more recent works of the named researchers also focus on maximizing the gap between two adjacent eigenfrequencies but using other methods of topology optimization [7]. In [8,9] topology optimization problems were solved by an iterative procedure based on a gradient-based algorithm.

Methods of solutions for band-gap problems also include genetic algorithms (GAs) and other global heuristic approaches [10]. However, most of the work in this area applies to topology optimization in terms of wave propagation [11,12] or phononic crystals and metamaterials optimization [13,14,15,16]. Therefore, optimization mainly concerns the material structure and not optimization by distribution of isotropic materials in structural elements with specific support conditions. In [17] the authors maximized the band-gap size for bending waves in an infinite periodic Mindlin plate. They also constructed a finite periodic plate using a number of optimized base cells in a postprocessed version. Band-gap optimization of infinite structures is also found in [18], where the authors used a generalized gradient ascent method for an optimization procedure. Shen et al. [19] developed a two-stage GA with a floating mutation probability to design a two-dimensional (2D) PnC of a square lattice with a maximal absolute band gap. A similar topic was discussed in [20], where Dong et al. maximized the relative widths of the gaps between adjacent energy bands of a 2D PnC with a square lattice using the finite element method (FEM) and a two-stage GA. PnC unit cell optimization with GAs was also described in [21], where a thin plate composed of aluminum and epoxy resin was presented.

In the area of dynamics and vibrations, genetic algorithms are used for numerous optimization problems. Biswal et al. [22] used GAs with FEM for the design and analysis of a nonprismatic piezolaminated cantilever beam for optimal vibration energy harvesting. GAs were also used for band-gap problems in periodic structures. Shi et al. [23] considered the inverse problem of a flexural vibrational band gap of a periodically supported beam. Further GA optimization problems were presented in [24,25]. The former concerns the frequency response for a locally resonant metamaterial beam, and the latter presents a design method to optimize the material distribution of functionally graded beams with respect to vibration and acoustic properties. GAs were also used to change the shape of a mechanical system under a nonlinear response for a clamped–clamped beam [26]. In [27] authors used GAs to solve a structural damage identification problem for railway bridges. The fitness function for this problem was defined based on the dynamic response of a bridge under a trainload. Structural damage identification in engineering structures was also considered in [28], but the subjects of the research were spatial Timoshenko arches. The authors used GAs to minimize the difference between damage properties and measured modal quantities for the structures.

Genetic algorithms are also used in problems directly related to structural engineering, but under strength conditions, to minimize material use, and consequently the overall cost of the structural element. In [29] authors used GAs as an auxiliary tool for pre-dimensioning prestressed concrete beams. They optimized the cross-section of an I-beam to develop one with the lowest cost of manufacturing that meets SLS and ULS conditions. An attempt to optimize the cost of a prestressed concrete element was also presented by Aydın and Ayvaz [30]. They determined the optimum span number and optimum cross-sectional properties of multi-span bridges using a hybrid GA. Similar tools were also used to optimize structures of roofs like domes [31] or steel trusses [32,33].

The goal of this work is to find the optimal thickness distribution of a linearly elastic beam. The design variable is the height of its cross-section, keeping its length and width constant. Natural frequencies of the beams are computed using the finite element method. An optimizer based on a genetic algorithm is used to determine the optimal material distribution in terms of maximizing the relative gap between two neighboring frequencies.

The paper is outlined as follows. The theoretical background and fundamental equations as well as the methodology (finite element method and genetic algorithms) are presented in Section 2. Section 3 is a description of the research problem and studied cases. Discussion of the optimization process using genetic algorithms is briefly presented in Section 4. In Section 5, results analysis and discussion are provided. The paper ends with general conclusions.

## 2. Fundamental Equations and Methods

### 2.1. Euler–Bernoulli Beam Theory

The analyzed element is a beam made of isotropic linear-elastic material of Young’s modulus *E*, Poisson number *ν*, and density *ρ*. The geometry of the beam of length *L* is described in an orthogonal Cartesian coordinate system *Oxyz*. The *Ox* axis coincides with the axis of the beam (Figure 1). The cross-section of the beam is a rectangle with dimensions *b* × *h*(*x*), which are parallel respective to the *Oy* and *Oz* axes. The beam is represented by a model according to the Euler–Bernoulli beam theory, which assumes that the segment which is straight and perpendicular to the beam’s *Ox* axis before deformation remains straight and perpendicular after deformation. In this chapter variational formulation of the fundamental equations for this problem is briefly introduced.

Let EJ=E(x)J(x) be flexural stiffness, μ=ρ(x)A(x) be mass per unit, w=w(x,t) be transverse deflection, q=q(x,t) be the transverse load, and c=c(x) be the damping coefficient. Therefore, strain and kinetic densities per unit of the beam are defined as follows:
(1)$$W=\frac{1}{2}EJ{\left(\frac{\partial^{2}w}{\partial x^{2}}\right)}^{2},K=\frac{1}{2}\mu {\dot{w}}^{2}.$$

The overdot stands for the derivative with respect to time $t=t_{0},t_{1}$. In the presented problem, the strain and kinetic energies are integral forms of values defined in (1), and the strain energy refers to beam bending energy.

The action functional is given by:(2)$$A=\int_{t_{0}}^{t_{1}}\int_{0}^{L}Ldxdt,$$
where *V* is the volume, and the Langragian is of the form:(3)L=K−W−pw+qw.

The dissipative force is assumed to be as follows:(4)$$p=p(x,t)=c(x)\dot{w}(x,t).$$

The equations of motion can be obtained from the extended (cf. [34]) principle of stationary action, formulated as:(5)$$\delta A=\int_{t_{0}}^{t_{1}}\int_{0}^{L}\delta Ldxdt=0,$$

After applying Equations (2) and (3) to (5) and performing integration by parts, the following formula was obtained:(6)$$\delta A=\int_{t_{0}}^{t_{1}}\int_{0}^{L}\left[\mu \ddot{w}\delta w-EJ\delta {\left(\frac{\partial^{2}w}{\partial x^{2}}\right)}^{2}-(q-p)\delta w\right]dxdt=0.$$

Formula (6) can also be rewritten and presented as (7). It constitutes the governing equation of the Euler–Bernoulli beam model:(7)$$\frac{\partial^{2}}{\partial x^{2}}\left(EJ\frac{\partial^{2}w}{\partial x^{2}}\right)+c\ddot{w}+\mu \ddot{w}=q.$$

Omitting the terms involving external load and damping, we arrive at an equation describing the natural bending vibrations of a beam:(8)$$\frac{\partial^{2}}{\partial x^{2}}\left(EJ\frac{\partial^{2}w}{\partial x^{2}}\right)+\mu \ddot{w}=0,$$
from which natural frequencies and mode shapes can be calculated, depending on the boundary conditions.

### 2.2. Dynamics in the Finite Element Method Approach

In problems of dynamics and free vibrations described by differential equations, the finite element method (FEM) is used. The generalized equation of motion is given as:(9)$$M\ddot{q}(t)+C\dot{q}(t)+Kq(t)=F(t),$$
where **M** is mass matrix, **K** is stiffness matrix, **C** is damping matrix, **F** is the external force vector, and $q(t),\dot{q}(t),\ddot{q}(t)$ are displacement, velocity, and acceleration vectors, respectively. The matrices of mass, stiffness, and damping for the entire structure can be defined as follows (the superscript *e* denotes single finite element *e*):(10)$$\begin{array}{c} M=\sum_{e}\int_{V^{\left(e\right)}}\rho^{\left(e\right)}N^{T(e)}N^{\left(e\right)}dV^{\left(e\right)}=\sum_{e}M^{\left(e\right)},\\ K=\sum_{e}\int_{V^{\left(e\right)}}B^{T(e)}D^{\left(e\right)}B^{\left(e\right)}dV^{\left(e\right)}=\sum_{e}K^{\left(e\right)},\\ C=\sum_{e}\int_{V^{\left(e\right)}}\mu^{\left(e\right)}N^{T(e)}N^{\left(e\right)}dV^{\left(e\right)}=\sum_{e}C^{\left(e\right)}. \end{array}$$
**B**^(*e*)^ stands for linear strain matrix of elasticity of element *e*, **D**^(*e*)^ is the elasticity matrix of element *e*, **N**^(*e*)^ is the shape function vector of element *e*, and *μ*^(*e*)^ is the damping parameter of element *e*. Here, the damping matrix was defined according to the Rayleigh damping model, commonly used in numerous studies [28,35]:(11)C=αM+βK.

Rayleigh coefficients *α* and *β* were assumed to be 2 × 10^−6^ and 3 × 10^−5^, respectively.

In the case of forced harmonic vibrations, the external force can be expressed as:(12)$$F(t)=F_{0}(t)\cos(\omega t).$$

Application of the multimodal approach for forced vibration analysis leads to the following solution:(13)$$q=a_{C}\cos\left(\omega t\right)+a_{S}\sin\left(\omega t\right),$$
where parameter *ω* is angular frequency and coefficients **a**_C_ and **a**_S_ are defined as:(14)$$\begin{array}{c} a_{C}={\left\{\left(K-\omega^{2}M\right)+\omega^{2}C\left[{\left(K-\omega^{2}M\right)}^{-1}C\right]\right\}}^{-1}F_{0},\\ a_{S}=\omega {\left(K-\omega^{2}M\right)}^{-1}C{\left\{\left(K-\omega^{2}M\right)+\omega^{2}C\left[{\left(K-\omega^{2}M\right)}^{-1}C\right]\right\}}^{-1}F_{0}. \end{array}$$

In this particular optimization problem, since we consider free vibrations, the damping matrix is equal to zero and no external forces are involved. Thus, Equation (8) takes the form of:(15)$$M\ddot{q}(t)+Kq(t)=0.$$

The solution and its second time derivative can be expressed as:(16)$$q=q_{a}\sin(\omega t),$$
where **q**_a_ stands for eigenvector. It can be expressed as follows, where *m* is number of degrees of freedom:(17)$$q_{a}=\left[q_{a,1},q_{a,2},\ldots ,q_{a,m}\right].$$

Assuming that eigenvalue *λ* is equal to *ω*^2^, Equation (13) can be written as:(18)$$\left(K-\lambda M\right)q_{a}=0.$$

Only the result where **q**_a_ ≠ 0 is considered, so in order to determine eigenvalues and eigenvectors, the following condition has to be met:(19)$$\det\left|K-\lambda M\right|=0.$$

### 2.3. Genetic Algorithms

Genetic algorithms are optimization techniques inspired by the principles of Darwin’s theory of natural selection and genetics. They are probabilistic algorithms that maintain a population of potential solutions. The solutions are subjected to genetic operations such as selection, crossover, and mutation that mimic the process of biological evolution. In this section genetic operators and strategies are explained according to the following works [36,37,38,39]. The algorithms used in this work have been adapted to suit the requirements of the investigated problem. They were created using the Python programming language supported mainly by the following libraries: NumPy, SciPy, and Matplotlib.

The beam was divided into *n*_e_ elements, each of which is of height *h_i_* and width *b*, where *i* = 1, 2, …, *n*_e_. Each beam element is a single gene—the basic parameter (a variable). The genes are represented by values corresponding to the height of elements *h_i_* and are joined into a string to form a chromosome. In the presented work, the chromosomes correspond to the individual input beams that constitute a population. Figure 2 illustrates the basic concepts behind optimization using the presented GA.

Evolution starts with an initial population (population ‘0’). In this study it consists of *n*_p_ individuals (chromosomes). Population ‘0’ is randomly generated from a beta distribution and it is represented by matrix **H** with dimensions *n*_e_ × *n*_p_. The values of **H** are samples from the beta distribution limited by the minimum and maximum sizes of single element heights *h*_min_ = 0.5 *h_i_* and *h*_max_ = 2 *h_i_*, where *h_i_* = 10^−2^ m. The resulting values in **H** represent a random population of individual solutions for the optimization problem. Each column represents an individual, and each row represents an element with a given position.

The next step is the evaluation of a created population. Population evaluation is performed in terms of the objective function which calculates relative difference between two adjacent frequencies. In this study, the objective function is defined as follows:(20)$$\max\Delta \omega_{k}=\max\frac{\omega_{k+1}-\omega_{k}}{\omega_{k+1}},$$
where *ω* is natural frequency of optimized beams, and index *k* represents the number of natural vibration frequency. The function describes the absolute width of the band gap. Adapting dimensionless values with a maximum value of 1 is advantageous and convenient in terms of numerical calculations.

To evaluate the created population, it is necessary to determine the natural vibration frequencies of each individual. Calculations were performed using our own FEM procedure (described in the previous subsection) built into the algorithm. The individuals of the initial population were assessed in terms of meeting the condition of the objective function maximalization and the best were then subjected to the genetic operators.

One of the used operators is mutation. It is applied to all individuals except the best one among the remaining individuals. The mutation used in the presented algorithm involves randomly selecting one gene (element) from the chromosome (beam) and modifying its value based on the chosen mutation type—addition or subtraction of the specified value to the initial gene.

Then, the algorithm selects parents using the roulette wheel selection mechanism. It selects parents until their number reaches the desired value (*n*_par_). The parent selection algorithm ensures that a certain number of parents *n*_par_ is selected for crossover by sampling from the cumulative probabilities. The first half of the selected parents is chosen based on their probabilities, while the second half is randomly chosen from the entire population, to avoid premature convergence. This can happen when similar chromosomes become dominant in a population. The algorithm then generates all possible pairs of selected parents for crossover using permutations. Value *n*_par_ is determined using a mathematical equation based on the population size, the number of remaining individuals, and the number of obtained offspring, to ensure a constant population size.

Offspring are created through crossover operations. The algorithm randomly chooses between one-point crossover (OPCX), blend crossover (BLXa), or two-point crossover (TPCX) based on predefined probabilities. OPCX operates on two parents A and B, and selects a random point, where the chromosome separates. The first child is created by concatenating the first part of parent A with the second part of parent B. Similarly, the second child is created by concatenating the first part of B with the second part of A. BLXa also combines two parents A and B. It randomly generates new individuals by blending the genes within a specified range based on a defined blending factor. The last one, TPCX, performs a two-point crossover between two parents. It selects two random crossover points and creates two children by combining the values before the first crossover point, between the two points, and after the second crossover point.

The resulting offspring are added to population **H**. Population size is adjusted to a selected value by removing any extra individuals created during crossover operations. Finally, the updated population **H** is returned by the algorithm, and the described procedure is repeated.

In general, genetic algorithms are not classified as “global optimizers” [40] and it can be observed in many optimization problems that GAs show a tendency to converge towards local optima. The likelihood of occurrence of this feature depends on the definition of the fitness function. There are methods to reduce this problem, e.g., by using different fitness functions, increasing the rate of mutation, or by using selection techniques that maintain diversity in the population. However, there are no general solutions to this problem. In this paper several techniques were employed to maintain population diversity (the aforementioned mutations and crossovers), but there is no guarantee that they always provide solutions in the form of global extremes.

## 3. Problem Statement

We consider a beam of Young’s modulus E = 205 GPa, Poisson number ν = 0.3, and density *ρ* = 7850 kg/m^3^. It is assumed that the length of the beam is *L* = 1 m.

Two types of beam boundary conditions were considered (Figure 3): simply supported at both ends (SS) and clamped-free (CF). Individuals were divided into *n*_e_ elements—32 or 64. The number of individuals in each population was assumed to be 320. The number of created populations (*n*_pop_) corresponds with the number of elements—200 for *n*_e_ = 32 and 300 for *n*_e_ = 64. Gaps between adjacent vibration frequencies were investigated in the range of *k* from 1 to 8. Moreover, all cases were performed three times, and they differed only in automatically generated (random variable) parameters, to compare the influence on GA solutions. As a result, the total number of cases subjected to the optimization procedure, and then analyzed, is 96.

First, the presented genetic algorithm with an integrated FEM procedure for eigenvalue problems was used. The number of finite elements in the FEM procedure was equal to the number of genes for each individual *n*. Depending on the boundary conditions (BC or SS), a number of dynamic degrees of freedom (DOF) was defined, assuming that for each node it could be up to three.

In addition, forced vibration analysis was performed for optimized individuals. The maximum displacement *w*_max_ was analyzed for both boundary condition cases. The excitation force *F*(*t*) = *F*_0_(*t*) cos(*ωt*) is located at the node closest to the support. Deflection was investigated at the mid-span (for an SS beam) or at the end of the cantilever (for a CF beam). The described dynamic problem is presented in Figure 4.

The obtained results (optimized individuals) were compared with reference individuals. The reference individual is a prismatic beam of rectangular cross-section *b* × *h*_ref_, where *h*_ref_ was calculated as the average height of the optimized individuals. As a result, both individuals—reference and optimized—have the same volume, but different material distribution.

## 4. Optimization Process Discussion

Here, examples of the optimization process in terms of eigenvalue problems are presented. The results are presented in plots only for selected cases as examples and briefly described. The examples in Figure 5, Figure 6 and Figure 7 align with selected cases in Figure 8, Figure 9 and Figure 10 (Figure 5 with Figure 10d, Figure 6 with Figure 9d, and Figure 7 with Figure 8a), to emphasis how the values obtained from GA optimization (individual element sizes) correspond to the resulting beams.

Population “0” in each case consists of individuals with completely random genes, arranged without any specific order. Initial signs of individuals’ fitness to the defined objective function in this problem start to appear as early as the 3rd or 4th population (for a division into *n*_e_ = 32 elements) or the 6th or 7th population (for a division into *n*_e_ = 64 elements). By the 10th population, specific areas can be observed where each individual experiences a decrease or increase in thickness. With each subsequent population, these areas are more and more clearly marked. The evolution process for example cases is selectively presented in Figure 5, Figure 6 and Figure 7, where elements of each beam are marked on the vertical axis, and the number of each individual (beam) in the population is presented on the horizontal axis.

From the populations numbering about 150 (for *n*_e_ = 32) and 200 (for *n*_e_ = 64), the changes between individuals within a particular population as well as changes between neighboring populations are very slight. As a result, the best individuals selected from successive populations no longer differ significantly in terms of matching the fitness function. However, it can be noted that convergence is slowest for elements near the ends of the beam.

To investigate the influence of the number of elements *n* on the evolution and final optimization result, additional plots, presented in Figure 8, Figure 9 and Figure 10, were created. The largest differences are observed in the initial populations, reaching up to approximately 10% in the maximum objective function values achieved by the best individual. In the final populations, these differences decrease below 5%. This trend persists when the optimization is focused on values Δ*ω*_1_ to Δ*ω*_6_, corresponding to lower natural frequencies. However, for frequencies Δ*ω*_7_ and Δ*ω*_8_, these differences are higher—up to several percent.

Furthermore, individuals divided into *n*_e_ = 32 elements achieve slightly better results in the optimization process for differences in frequencies Δ*ω*_1_ and Δ*ω*_2_. On the other hand, for optimization targeting the highest frequencies Δ*ω*_7_ and Δ*ω*_8_, a division into *n*_e_ = 64 elements appears to be more advantageous. These observations apply to both SS and CF beams (Figure 11).

For each optimization case, three attempts were performed—marked as ‘take 0’, ‘take 1’, and ‘take 2’—to compare random variable parameters’ influence on optimization algorithm results. In most cases the results do not differ significantly, up to 5% between the different approaches (Figure 12a,b). In individual cases, the final optimization result is slightly different (around 7–8%, Figure 11c), but the results still exhibit extremely strong convergence.

## 5. Results

### 5.1. Analysis in Terms of Eigenvalue Problems

In this section, an analysis of obtained results and sample results from the optimization process are presented. Firstly, matching the winning individuals to the objective function will be considered. Here, the best individuals from every population, for every considered case, are presented: SS in Figure 13 and Figure 14, and CF in Figure 15 and Figure 16.

The results presented in the following subsection provide further insight into the analysis regarding the influence of the number of elements *n* on the quality of the obtained optimum solution.

For both SS and CF beams there are no significant differences between cross-section distribution of obtained best solutions for *n*_e_ = 32 and 64, but only at low frequencies (Δ*ω*_1_ and Δ*ω*_2_). As the natural frequency mode number increases, differences in the cross-sectional shapes of the beams also increase, and consequently the differences in their fitness function values also increase. Exact values of the differences are presented in Table 1. For Δ*ω*_1_ the differences remain within the range of 0.05% or less, for both SS and CF. At higher frequencies, the increase in differences depends on the static scheme and in most cases is within 5%. The highest values of difference were observed for Δ*ω*_8_.

Figure 17 illustrates the relationship between the optimized beam cross-section and mode shapes. Clear dependencies can be observed between the locations of cross-sections with reduced mass and stiffness, and the shape of the natural mode. In places where the higher frequency (*ω_k_*_+1_) mode shape plot crosses the zero-amplitude value (which are also its inflection points), and simultaneously the lower frequency mode (*ω_k_*) shape plot reaches a local extreme, a reduction in mass and stiffness occurs. This indicates that there is a correlation between the locations of mass and stiffness reduction and the characteristics of the mode shapes. This regularity applies to both SS and CF beams, but only for areas inside the span.

Comparison of the natural frequencies between several examples of reference and optimized beams is presented in Figure 18, Figure 19, Figure 20 and Figure 21. Both for SS and CF beams, the optimized individuals obtain the best results compared to the reference beams at higher eigenfrequencies. Furthermore, slightly better results are achieved by individuals divided into 64 elements compared to those divided into 32 elements. In the case of Δ*ω*_4_, the optimized beam is twice as good as the reference individual, and for Δ*ω*_8_, it is even three times better. For differences in the lowest natural frequencies, improvements in the properties of the optimized beam are also evident, although they are modest, around 10–15% for Δ*ω*_1_. The comparison results are similar for SS and CF beams. The results are also presented in Table 2 and Table 3.

### 5.2. Analysis in Terms of Forced Vibrations

Here, we present analysis of optimized beams in terms of dynamic properties. The optimization aimed at maximizing gaps between adjacent frequencies also affected their dynamic amplitude response properties. The maximum amplitude of the beams was analyzed by comparing the optimized individuals with the reference individuals. The results are presented in Figure 22, Figure 23, Figure 24 and Figure 25.

Although the optimization process undeniably influenced the resonance response, it is not possible to define a clear relationship describing this impact. For SS beams, in the majority of analyzed cases, the maximum amplitude increased after optimization, but only in the low-frequency range. For CF beams, the maximum deflection was smaller for optimized individuals, but only when optimization was conducted within the Δ*ω*_1_–Δ*ω*_4_ range. For Δ*ω*_5_–Δ*ω*_8_, the optimized individuals achieved either higher or lower maximum amplitude values. It was also noted that the value of *n*_e_ had a significant influence on the amplitude response. In some cases, depending on the element division, the same beam achieved a higher amplitude after or before optimization.

## 6. Summary and Conclusions

In the presented paper, application of genetic algorithms for the optimization of simply supported and clamped-free beams was investigated, with a focus on maximizing the gaps between adjacent natural frequencies. The obtained results were analyzed to formulate the following conclusions:The application of GAs proved to be effective in optimizing the beams for maximizing the gaps between natural frequencies. Optimized beams exhibited increased gaps between adjacent natural frequencies, according to the defined fitness function.Based on the conducted analyses, it can be concluded that for relatively low natural frequencies (Δ*ω*_1_, Δ*ω*_2_), a division into larger elements can be successfully applied. This will result in shorter computational time while maintaining satisfying results.The randomness of parameters in the GA has a negligible influence on the final results. This was confirmed by the analysis of three independent approaches for each case. The ultimate results obtained from each approach for individual cases do not significantly differ.All optimized beams exhibit a periodic-like structure that is strongly correlated with the mode shape. Mass and stiffness reduction occurs at points where the lower mode shape of adjacent frequency *k* crosses the amplitude axis at 0 and the higher mode shape of frequency *k* + 1 reaches an extreme value.Both SS and CF optimized beams performed better than the reference beams at higher vibration frequencies.Optimization aimed at improving the properties of beams related to natural vibrations also resulted in a dynamic response. The optimization process had a complex impact on the resonance response, with no clear relationship identified.Although the adopted objective function is convenient to use due to limited numerical values, it may, however, lead to solutions in which the absolute value of the gap is not the maximum.

It has to be stressed that in GA algorithms, as a heuristic, stochastic search method, there is no proof that the solution is the global extreme. However, obtained results suggest that GAs can be a valuable tool for engineers and researchers in optimizing beams with specific frequency requirements.

Further research could be directed into applying more accurate beam models, such as Timoshenko–Ehrenfest thick beam theory, the use of finite elements with variable stiffness, and imposing restrictions on differences in the dimensions of adjacent elements. It would also be desirable to obtain experimental results for dynamics characteristics of optimized elements.

## Figures and Tables

**Figure 1 materials-16-04963-f001:**
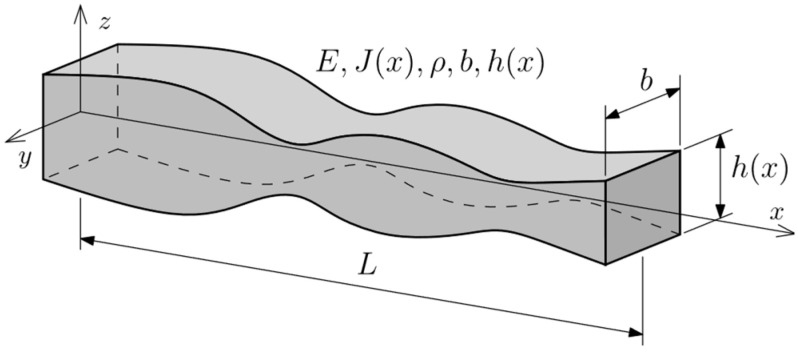
Computational model of the beam.

**Figure 2 materials-16-04963-f002:**
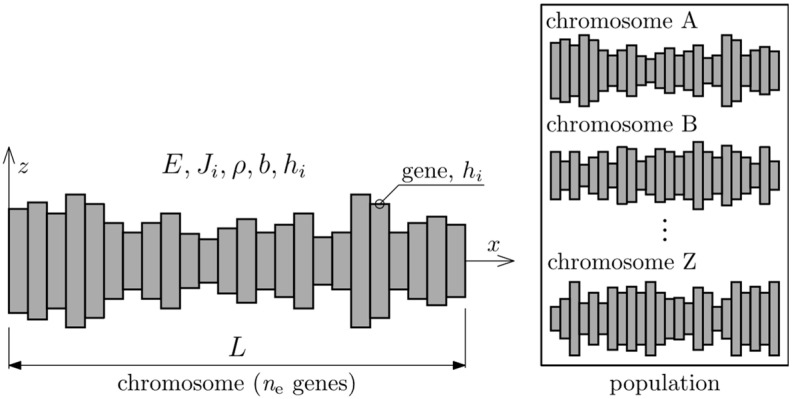
Graphic representation of GA basic elements for presented problem.

**Figure 3 materials-16-04963-f003:**
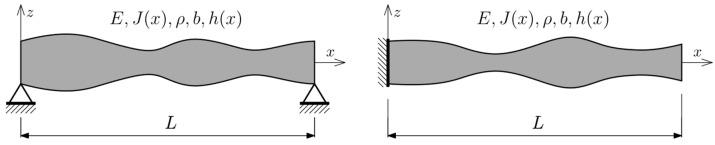
Beam boundary conditions: **left**—simply supported; **right**—clamped-free.

**Figure 4 materials-16-04963-f004:**
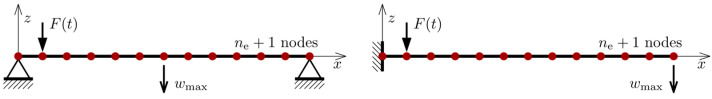
Forced vibration analysis: **left**—SS beam; **right**—CF beam.

**Figure 5 materials-16-04963-f005:**
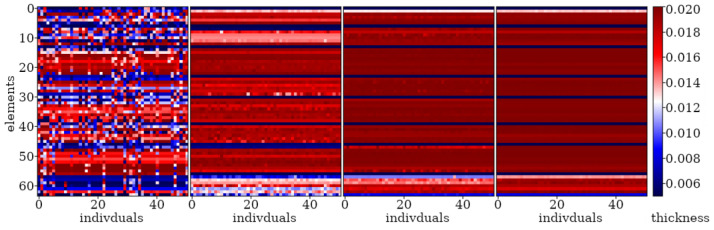
Evolution of the example population; from left to right: population no 10, 100, 200, and 290. Optimization in terms of Δ*ω*_8_ for CF beam, *n*_e_ = 64.

**Figure 6 materials-16-04963-f006:**
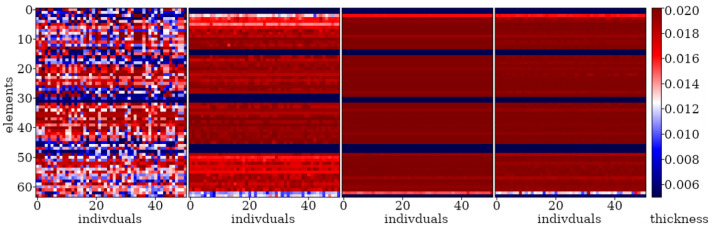
Evolution of the example population; from left to right: population no 10, 100, 200, and 290. Optimization in terms of Δ*ω*_4_ for CF beam, *n*_e_ = 64.

**Figure 7 materials-16-04963-f007:**

Evolution of the example population; from left to right: population no 10, 100, and 190. Optimization in terms of Δ*ω*_1_ for SS beam, *n*_e_ = 32.

**Figure 8 materials-16-04963-f008:**
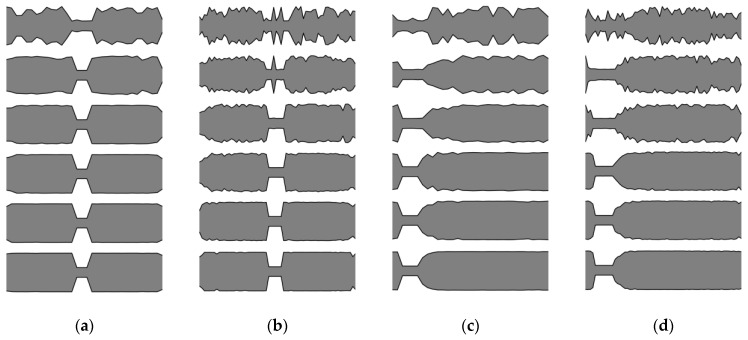
Evolution of the best individuals from the example populations (starting with 10th population at the top); optimization for Δ*ω*_1_: (**a**) SS *n*_e_ = 32; (**b**) SS *n*_e_ = 64; (**c**) CF *n*_e_ = 32; (**d**) CF *n*_e_ = 64.

**Figure 9 materials-16-04963-f009:**
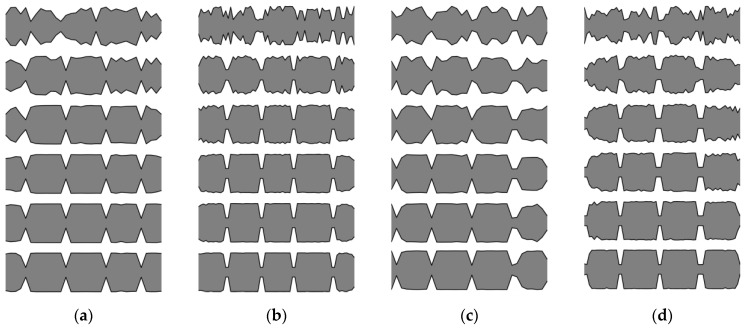
Evolution of the best individuals from the example populations (starting with 10th population at the top); optimization for Δ*ω*_4_: (**a**) SS *n*_e_ = 32; (**b**) SS *n*_e_ = 64; (**c**) CF *n*_e_ = 32; (**d**) CF *n*_e_ = 64.

**Figure 10 materials-16-04963-f010:**
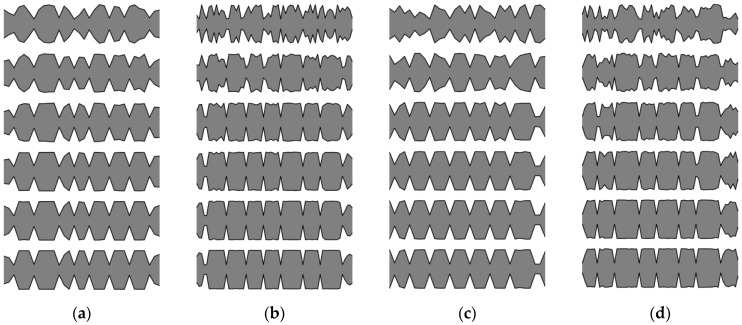
Evolution of the best individuals from the example populations (starting with 10th population at the top); optimization for Δ*ω*_8_: (**a**) SS *n*_e_ = 32; (**b**) SS *n*_e_ = 64; (**c**) CF *n*_e_ = 32; (**d**) CF *n*_e_ = 64.

**Figure 11 materials-16-04963-f011:**
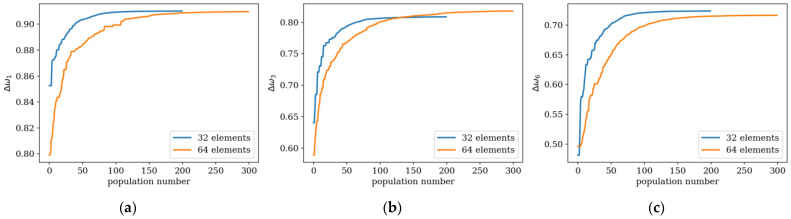
Values of the fitness function for the best individuals depending on the population number for *n*_e_ = 32 and 64 elements; optimization in terms of (**a**) Δ*ω*_1_ for SS beam; (**b**) Δ*ω*_3_ for CF beam; (**c**) Δ*ω*_6_ for SS beam.

**Figure 12 materials-16-04963-f012:**
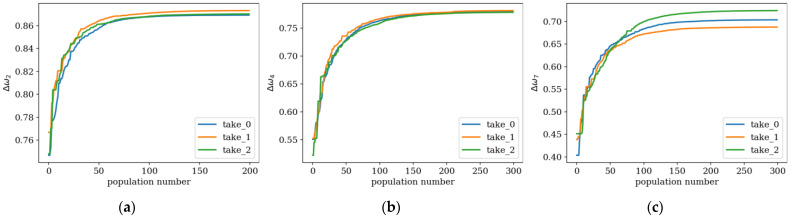
Values of the fitness function for the best individuals from three trials; optimization in terms of (**a**) Δ*ω*_2_ for CF beam, *n*_e_ = 32; (**b**) Δ*ω*_4_ for CF beam, *n*_e_ = 64; (**c**) Δ*ω*_7_ for SS beam, *n*_e_ = 64.

**Figure 13 materials-16-04963-f013:**
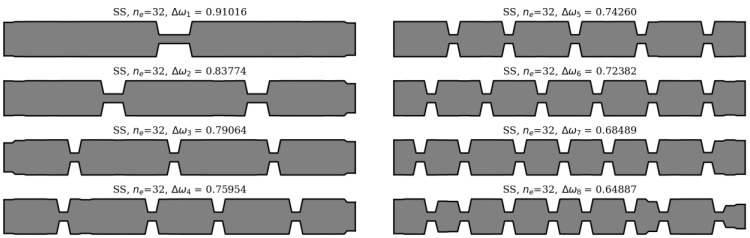
The winning individuals for SS beam, *n*_e_ = 32.

**Figure 14 materials-16-04963-f014:**
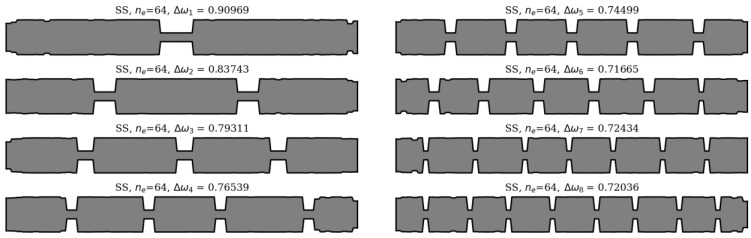
The winning individuals for SS beam, *n*_e_ = 64.

**Figure 15 materials-16-04963-f015:**
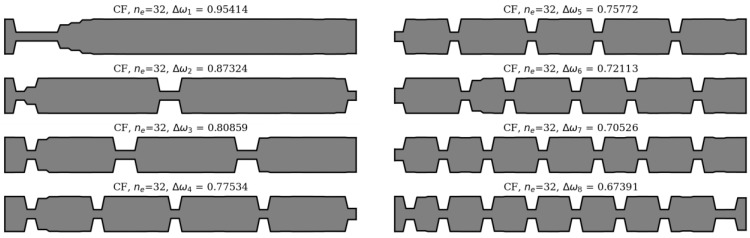
The winning individuals for CF beam, *n*_e_ = 32.

**Figure 16 materials-16-04963-f016:**
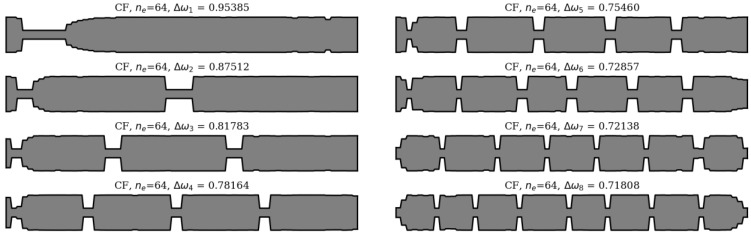
The winning individuals for CF beam, *n*_e_ = 64.

**Figure 17 materials-16-04963-f017:**
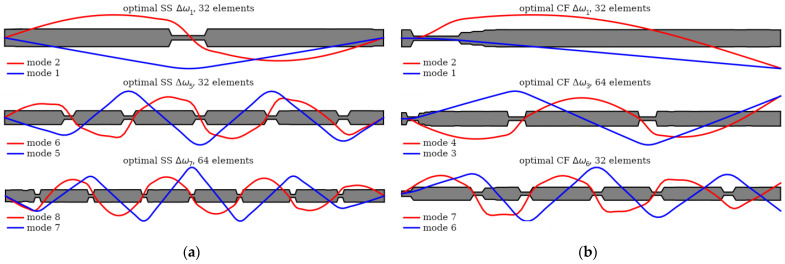
The relationship between the optimized beam cross-section and the mode shapes for chosen (**a**) SS beams and (**b**) CF beams.

**Figure 18 materials-16-04963-f018:**
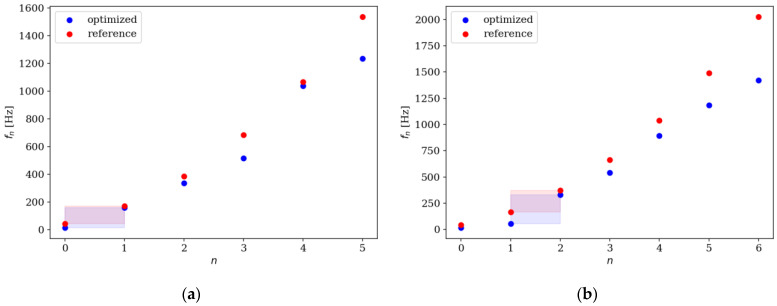
Comparison of the natural frequencies of reference and optimized beams for SS beam *n*_e_ = 32: (**a**) Δ*ω*_1_; (**b**) Δ*ω*_2_.

**Figure 19 materials-16-04963-f019:**
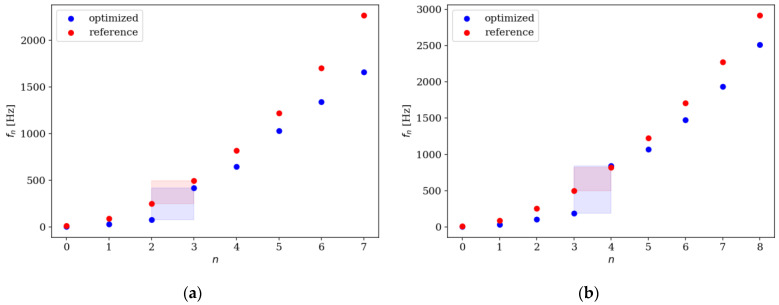
Comparison of the natural frequencies of reference and optimized beams for CF beam *n*_e_ = 32: (**a**) Δ*ω*_3_; (**b**) Δ*ω*_4_.

**Figure 20 materials-16-04963-f020:**
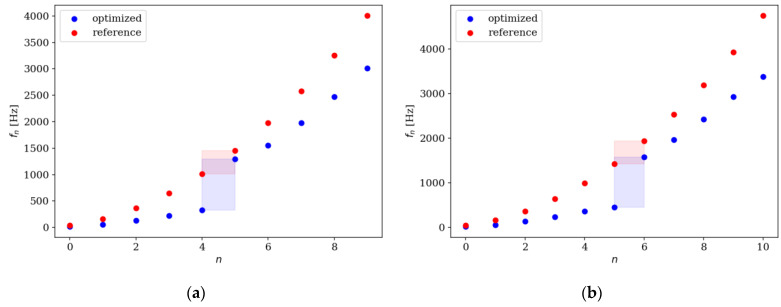
Comparison of the natural frequencies of reference and optimized beams for SS beam *n*_e_ = 64: (**a**) Δ*ω*_5_; (**b**) Δ*ω*_6_.

**Figure 21 materials-16-04963-f021:**
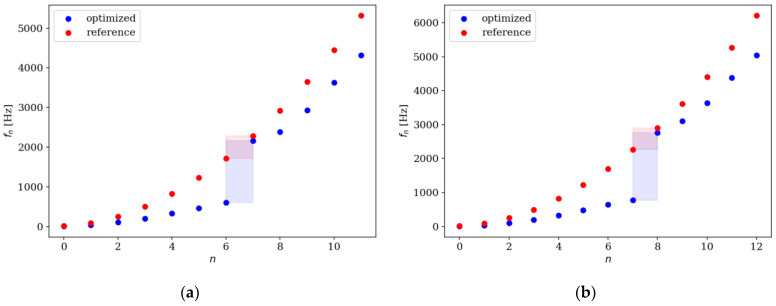
Comparison of the natural frequencies of reference and optimized beams for CF beam *n*_e_ = 64: (**a**) Δ*ω*_7_; (**b**) Δ*ω*_8_.

**Figure 22 materials-16-04963-f022:**
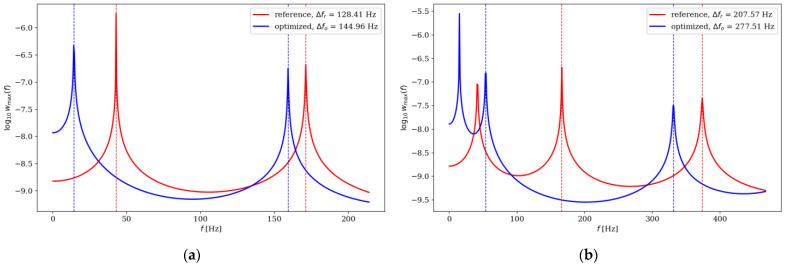
Comparison of the amplitude response of reference and optimized beams for SS beam *n*_e_ = 32: (**a**) Δ*ω*_1_; (**b**) Δ*ω*_2_.

**Figure 23 materials-16-04963-f023:**
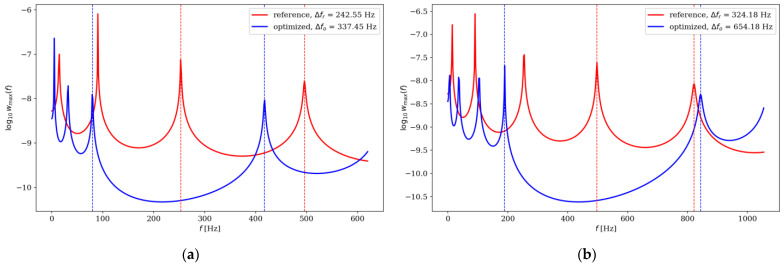
Comparison of the amplitude response of reference and optimized beams for CF beam *n*_e_ = 32: (**a**) Δ*ω*_3_; (**b**) Δ*ω*_4_.

**Figure 24 materials-16-04963-f024:**
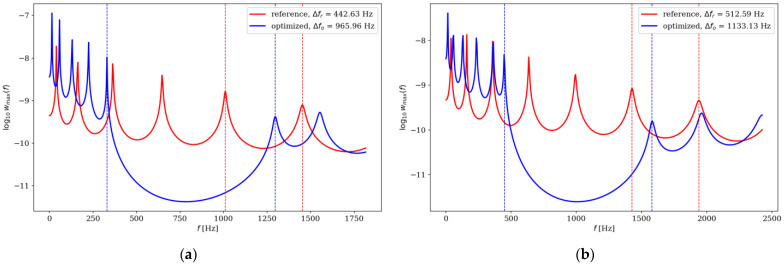
Comparison of the amplitude response of reference and optimized beams for SS beam *n*_e_ = 64: (**a**) Δ*ω*_5_; (**b**) Δ*ω*_6_.

**Figure 25 materials-16-04963-f025:**
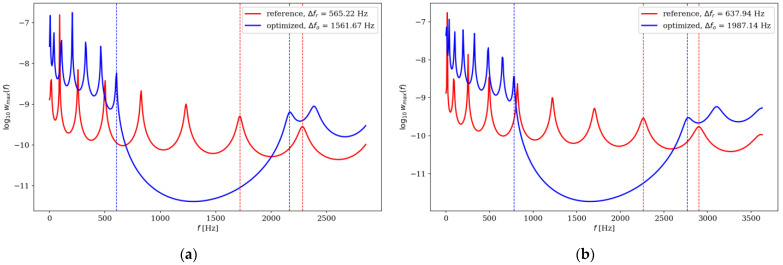
Comparison of the amplitude response of reference and optimized beams for CF beam *n*_e_ = 64: (**a**) Δ*ω*_7_; (**b**) Δ*ω*_8_.

**Table 1 materials-16-04963-t001:** The differences between objective function values between beams divided into *n*_e_ = 32 and *n*_e_ = 64 elements.

*k*	*n* _e_	SS Beam		CF Beam	
		Δ*ω_k_*	Difference, %	Δ*ω_k_*	Difference, %
1	32	0.91016	0.05	0.95414	0.03
	64	0.90969		0.95385	
2	32	0.83774	0.04	0.87324	0.21
	64	0.83743		0.87512	
3	32	0.79064	0.31	0.80859	1.13
	64	0.79311		0.81783	
4	32	0.75954	0.76	0.77534	0.81
	64	0.76539		0.78164	
5	32	0.74260	0.32	0.75772	0.41
	64	0.74499		0.75460	
6	32	0.72382	1.00	0.72113	1.02
	64	0.71665		0.72857	
7	32	0.68489	5.45	0.70526	2.23
	64	0.72434		0.72138	
8	32	0.64887	9.92	0.67391	6.15
	64	0.72036		0.71808	

**Table 2 materials-16-04963-t002:** Comparison of the natural frequencies of reference and optimized beams divided into *n*_e_ = 32 elements.

*k*	The Individual	SS Beam		CF Beam	
		Δ*f*	Difference, %	Δ*f*	Difference, %
1	reference	128.41	12.89	76.86	−11.10 *
	optimized	144.96		68.33	
2	reference	207.57	33.69	165.10	30.82
	optimized	277.51		215.99	
3	reference	296.76	73.45	242.55	39.17
	optimized	514.72		337.55	
4	reference	371.75	93.85	324.18	101.80
	optimized	720.62		654.18	
5	reference	445.22	108.18	403.84	96.85
	optimized	926.86		794.94	
6	reference	510.52	119.02	472.00	108.36
	optimized	1118.15		983.48	
7	reference	570.74	123.02	533.87	119.57
	optimized	1272.85		1172.22	
8	reference	603.70	137.95	574.00	130.03
	optimized	1436.51		1320.39	

* In the case of lower frequencies, it turns out that the absolute difference in frequencies can be smaller for beams optimized with respect to the defined fitness function.

**Table 3 materials-16-04963-t003:** Comparison of the natural frequencies of reference and optimized beams divided into *n*_e_ = 64 elements.

*k*	The Individual	SS Beam		CF Beam	
		Δ*f*	Difference, %	Δ*f*	Difference, %
1	reference	126.86	13.86	76.42	−8.75 *
	optimized	144.44		69.73	
2	reference	206.89	34.07	166.26	16.47
	optimized	277.38		193.65	
3	reference	285.49	62.88	247.30	47.54
	optimized	465.01		364.87	
4	reference	370.80	94.92	331.71	82.80
	optimized	722.76		606.35	
5	reference	442.63	118.23	405.86	101.58
	optimized	965.96		818.15	
6	reference	512.59	121.06	475.61	127.38
	optimized	1133.13		1081.46	
7	reference	622.68	254.00	565.22	176.29
	optimized	1850.31		1561.67	
8	reference	691.80	217.97	637.94	211.49
	optimized	2199.70		1987.14	

* In the case of lower frequencies, it turns out that the absolute difference in frequencies can be smaller for beams optimized with respect to the defined fitness function.

## Data Availability

The data presented in this study are available on request from the corresponding author.

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
