# Peer review of "Genetic Algorithm Optimization of Beams in Terms of Maximizing Gaps between Adjacent Frequencies"

_materials, 2023, doi:10.3390/ma16144963_

Round 1

Reviewer 1 Report

The authors present an interesting, so-called genetic algorithm to optimize/maximize the gap between the eigenfrequencies of a beam. While, there is still plenty room for the improvement of their presentation. The following is some examples.

1. About Eq. (1), the statement strain and kinetic energy are defined as follows. Firstly,  grammatically it should be : ““strain and kinetic energies are defined as follows. Secondly, the strain energy here is specially beam bending energy. Thirdly, when you call W and K as strain and kinetic energies, they are ambiguous because they are just the strain and kinetic energies of a differential element. And their corresponding integral forms are the strain and kinetic energies of a whole beam.

2. About Eq. (3), usually we define the Langragian as K-W as a convention, not W-K.

3. About the statement on Eq. (4): Applying stationary-action principle and divergence theorem leads to. For me, this is the application of the Hamiltonian Principle.

4. About the statement on Eq. (5):After applying equations (2), (3) to (4) and performing transformations, the following formula was obtained”. Here there is no transformation at all, it is simply the operation of the integration by parts. Furthermore, the differential equation of Eq. (6) is NOT the Euler-Lagrange equation! It is simply called the governing equation of the Euler-Bernoulli beam model.

5. From Eq. (7)-(12), the authors discuss the damping issue, which does not appear in their derivations and their result discussion. I suggest that the authors should simply delete them. Furthermore, their statement on the underdamping as presented in Eq. (6) is wrong or at least is not rigorous.

6. I am confused with heir computation and discussion on the optimization/maximization on the eigenfrequencies. The key question here is as follows: Yes, their enlarge the eigenfrequency gaps by varying the beam thickness; but how do you know the results are the maximized ones? You need a proof or at least give me a straightforward illustration. And the maximum gaps as presented in the manuscript are the global maximum or local maximum?

7. The authors presented a very specific optimization computation and the dimensions and material properties are given in section 3. The thing is that a general result may appeal to much more people. For the Euler-Bernoulli beam, my suggestion is that you should use the nondimensionalization scheme and present the dimensionless results. As the researchers in structural mechanics, you should know that researchers actually are more familiar with the (dimensionless) numbers such 1.875 and 4.73 etc.

8. Section 5 and other places. Mathematically there is no term called eigenproblem, which is given in Eq. (17). It is called eigenvalue problem.

9. Actually, my favourite results are those presented in Figs. 22-25. Although those amplitude responses are the simple and straightforward results once you determine/optimize the eigenfrequencies of a beam. You should say more on their potential applications. 

The English presentation itself is OK. To my surprise, there is a significant room for the improvement of  the authors' use of the scientific terminology.

Reviewer 2 Report

The presented paper focus on optimizing the gaps between adjacent natural frequencies in a simply supported and clamped-free beam, using Genetic Algorithms.

The study process is adequately described and the use of such an algorithm is justified. Several study cases have been considered, and clearly presented. 

The reviewer suggests some improvements to the paper.

1. The minimization of Δf as the objective function is interesting, and the results based on this function deliver the intention of the authors, however, the theoretical background of this function is not completely clear. It is advised to add a discussion around the idea of this function.

2. Figures 5 to 7 are not easy to understand,

2.1 Include a color bar to distinguish the meaning of the difference between red and blue.

2.2 clearly mention in the text that the entire beam structure is shown as elements in the Y-axis, and the X-axis shows the all beans suggested by GA in a particular iteration.

2.3 Show one example of the final shape of the mean, to understand the connection between the graph and the bean shape.

3. Offer an engineering meaning to the study of gap maximization, such as lightweight, vibrational characteristics tuning?

4. There exist several related studies that use such techniques for damage identification, the authors should show awareness of such studies.

good language
